# Effect of inverted visual acceleration profile on vestibular heading perception

**Miguel A. Yakouma**[1,2], **Eric Anson**[2,3], **Benjamin T. Crane** [1,2,3]*

**1** Department of Biomedical Engineering, University of Rochester, Rochester, New York, United States of America, **2** Department of Otolaryngology, University of Rochester, Rochester, New York, United States of America, **3** Department of Neuroscience, University of Rochester, Rochester, New York, United States of America

* craneb@gmail.com

## Abstract

Visual motion is ambiguous in that it can either represent object motion or self-motion. Visual-vestibular integration is most advantageous during self-motion. The current experiment tests the hypothesis that the visual motion needs to have a motion profile consistent with the inertial motion. To test this, we examined the effect on heading perception when the visual stimulus was consistent with the inertial motion compared to an inverted visual stimulus, which was thus inconsistent with inertial motion. Twenty healthy human subjects (mean age $20 \pm 3$ years, 13 female) experienced 2s of translation, which they reported as left or right. A synchronized 2s visual heading was offset by $0°, \pm45°, \pm60°,$ or $\pm75°$. In randomly interleaved trials, the visual motion was consistent with the inertial motion or inverted – it started at the peak velocity, decreased to zero mid-stimulus, and then accelerated back to the peak velocity at the end. When the velocity profile of the visual stimulus matched the velocity profile of inertial motion, the inertial stimulus was biased $10.0 \pm 1.8°$ (mean$\pm$SE) with a 45° visual offset, $8.9 \pm 1.7°$ with a 60° offset, and $9.3° \pm 2.5 \pm$ with a 75° offset. When the visual stimulus was inverted so it was inconsistent with the inertial motion, the respective biases were $6.5 \pm 1.5°, 5.6 \pm 1.7°,$ and $5.9 \pm 2.0°$. The biases with the inverted stimulus were significantly smaller ($p < 0.0001$), demonstrating that the visual motion profile is considered in multisensory integration rather than simple trajectory endpoints.

## Introduction

Multisensory integration should occur when two stimuli share a plausible common causation. Visual motion is ambiguous in that it can represent external object motion or self-motion through a fixed environment. For visual and inertial cues to be integrated, they must have self-motion as the common causation. Perception of heading direction is an ecologically relevant sensory-based ability in which visual and inertial

**Data availability statement:** All relevant data are within the paper and its Supporting Information files

**Funding:** Grant support was provided by 2 R01 DC013580. EA was supported in part by the National Institutes of Health (NIDCD K23 DC018303). The funders had no role in study design, data collection and analysis, decision to publish, or preparation of the manuscript.

**Competing interests:** The authors have declared that no competing interests exist.

cues are normally integrated [1–4]. How common causation is determined has been shown to depend on both spatial [5] and temporal alignment [6–10]. However, results to date have suggested that visual and inertial stimuli do not need consistent velocity and acceleration profiles [11]. The current study challenges that assumption by examining visual and inertial stimuli with very different motion profiles.

Close temporal alignment of the stimuli is a factor that is important, closer timing yields more robust multisensory integration [6–10]. This has been shown by our group: When visual and inertial stimuli are more than about 250 ms out of temporal alignment, minimal sensory integration occurs, and when they are more than 500 ms out of temporal alignment, they aren't integrated at all [12]. This happens despite significant temporal overlap between the visual and inertial stimuli with these offsets, suggesting that visual motion does not only need to occur during the inertial motion but also needs to have plausible common causation with self-motion based on the inertial stimulus. In these previous experiments, the peak velocity and acceleration were not modified independently of the beginning and end of the stimuli, so it is unclear if the most relevant factor was differences in onset time, end time, time of peak acceleration/velocity, or another factor.

When visual motion is consistent with self-motion, it is more likely to be interpreted as such. This has been studied in the context of visual stimuli presented alone where a more compelling sense of vection (i.e., the illusory perception of self-motion induced by a visual stimulus while the observer is stationary) occurs when there is an acceleration component and not just a constant velocity [13]. This is presumably because acceleration is more consistent with inertial motion. Furthermore, this study demonstrated that the sensation of vection becomes most compelling after the acceleration component subsides. Because the subject is stationary there was no inertial acceleration present, consistent with the vection sensation being dependent on its consistency with inertial motion. In a follow-up study further supporting the integration of visual acceleration, standing individuals reported consistently stronger vection when acceleration was superimposed on constant radial expansion or contraction [14].

Some degree of temporal alignment between visual and inertial motion should be required for visual-inertial multisensory integration [15], but which specific features of the stimuli need to be aligned are unclear. It seems likely that in addition to the start and end points of the visual and inertial motion being similar, the alignment of the velocity and/or acceleration profile could also be important. In a previous study in which subjects were asked to judge if a stimulus was straight ahead or offset, it was found that some subjects did not integrate the visual stimulus because it was thought to be directionally inconsistent with the inertial stimulus [16]. However, a subsequent study that varied the visual motion profile demonstrated that visual-inertial integration occurred equally well and in a statistically optimal manner when the velocity profile of the visual stimulus was either a constant velocity or when it matched the inertial stimulus [11]. The differences between these may be the much longer duration stimulus (around 10s) used in the earlier study. A subsequent study looked at the effect of stimulus duration and found that longer stimuli tended to weigh the visual component

of heading more [17]. These longer duration stimuli will have lower acceleration (closer to the perceptual threshold) thus, the inertial component likely had lower reliability, which could explain why the visual component had a higher influence on heading perception.

For the current study, we focus on 2s duration visual and inertial stimuli. This duration was used because it fits within the requirements of what can be comfortably and reliably delivered in our laboratory while avoiding potentially confounding velocity storage from longer movements. This is also the same duration that was used to previously show that temporal alignment is required [12] and somewhat paradoxically, a constant velocity visual stimulus is integrated the same as a velocity-matched visual stimulus [11], although in that study, the total duration was shorter (1s). The Butler, et al. study compared the fixed velocity visual stimulus with a velocity profile that matched the inertial stimulus; however, it did not try to vary the timing of the velocity and acceleration profile within the visual stimulus remained invariant.

The current study tests the hypothesis that features within the visual stimulus need to be consistent with the inertial stimulus for multisensory integration to occur. This was done by comparing the visual influence on inertial heading perception for stimuli where the motion profile of the visual and inertial stimuli are matched and when the visual stimulus velocity and acceleration were inverted. Both visual stimuli correspond to the same distance covered in space. The purpose of this study was not to look at causal inference [18] as it would have been obvious to most observers that the inverted visual stimulus was inconsistent with the inertial motion. We previously observed that visual stimuli bias inertial direction perception even with large offsets of up to 90° when it was clear to subjects that the stimuli were artificially offset [5]. Thus, the range of offsets used in the current study went to 75°. We hypothesized that there were three possible ways in which modification of the visual stimulus may influence multisensory integration: 1) It could narrow the angular deviation of visual and inertial stimuli in which integration could occur. This was tested by looking at ±45, ±60, and ±75° offsets between inertial and visual headings. If this were to occur it would imply that both heading disparity and velocity profile were considered in determining visual-inertial common causation. 2) The influence of the visual stimulus could be less. If this were the case the perceived heading direction would be closer to the inertial heading direction when the visual stimulus didn't match. 3) The integration could occur independently of the velocity profile of the visual stimulus, suggesting congruent path endpoints may be more relevant. In other words, the variation in the visual stimulus profile does not affect the inertia heading perception.

## Methods

### Ethics statement

The research was conducted according to the principles expressed in the Declaration of Helsinki. A written informed consent form was approved by the University of Rochester Research Science Review Board before the study was conducted. All subjects participated in the experiments between July and October of 2022.

### Human subjects

Twenty healthy subjects (13 female) were enrolled in the experiments. The mean age was 20±4 (mean±SD, the range was 18–31). The oldest subjects were #2 (31), and #20 (29). Eleven subjects were 18 years old. All subjects reported vision that was normal or corrected to normal.

### Equipment

A 6-degree-of-freedom (6-DOF) motion platform (Moog, East Aurora, NY model 6DOF2000E) was used to deliver the inertial motion stimuli. The setup is common in human motion perception studies [19] and has been described previously for heading estimation experiments [4,12,20,21]. A 55" color LCD screen with a 1920 x 1080 pixel resolution delivered the visual component. The screen was mounted to the motion platform and set 50 cm from the subject, filling a 117° horizontal

field-of-view. The subjects wore a helmet attached to the motion platform while they sat in an automotive-style racing seat. In this study, the head movement was not measured separately from the platform as significant decoupling was felt to be unlikely for the type of motion used. As detailed previously [22], an audible white noise was reproduced from two platform-mounted speakers on either side of the subject to mask sound from the platform.

## Stimuli

The visual component was formed by a star field in which each star was a 0.5 cm tall yellow triangle whose scale was adjusted at the plane of the screen to simulate distance as previously described [12]. A visual refresh rate of 60 Hz was used with 30% of points randomly repositioned in each frame (i.e., 70% visual coherence). This decreased coherence was used to encourage subjects to focus on the inertial stimulus. No fixation point was provided, the lights were switched off, and subjects faced the screen displaying the visual stimulus.

Acceleration-matched visual and inertial stimuli were both presented with a single 2s (0.5Hz) sine wave, in acceleration motion profile. The movement corresponded to 15 cm of displacement, which had a peak acceleration of 25 cm/s/s and peak velocity of 15 cm/s, which was more than an order of magnitude higher than human motion perception thresholds [23,24]. The inverted visual stimulus also lasted 2s (Fig 1) and had a similar peak acceleration (Fig 1A), peak velocity (Fig 1B), and displacement (Fig 1C). The difference between the normal and inverted visual stimulus was that the inverted stimulus began and ended at the peak velocity while the normal stimulus achieved the peak velocity at the midpoint (Fig 1B). However, both stimuli were consistent with the same displacement, and had the same peak velocity and peak acceleration.

Each trial block consisted of twelve randomly interleaved staircases. In half of these staircases, the inertial stimulus started 50° to the right, and in the other half it was 50° to the left. Both staircases could pass through zero in the opposite direction. At 50° offsets, all subjects were reliably able to identify the inertial heading direction as left or right, independent of the visual offset. Each of these six staircases in each heading direction were divided into three with an inverted visual stimulus and three with a normal visual stimulus. Within each set, there was one with the visual stimulus offset to the right, one aligned with the inertial stimulus (no offset), and one offset to the left. Each staircase included 15 stimulus presentations. Offsets of ±45, ±60, and ±75° were tested in different trial blocks in an order that was randomly determined for each subject. There was masking noise during each stimulus presentation. Subjects were asked to report the direction of the inertial stimulus as left or right of straight ahead. For staircases that started with a stimulus 50° to the left, after each leftward response, the next stimulus was shifted 8° to the right. After a rightward response, the step size was decreased by half (e.g., from 8° to 4°) and shifted to the left. With subsequent reversals stimuli step sizes could be reduced to a minimum of 1° or increased after three responses in the same direction. For the corresponding staircase that started to the right, there was a similar adjustment. Each staircase could step through zero. This resulted in the majority of stimuli late in the staircase being focused near the point of subjective equality (PSE) at which subjects were nearly equally likely to respond with left or right. If no direction was entered within 2s no response was recorded, and the stimulus was presented again the next time that staircase was active. These types of lapses were rare occurring less than 1% of the time.

## Analysis

The fraction of rightward responses for each stimulus level was plotted as a function of the heading direction tested for both the normal velocity (Fig 2A-2C) and inverted velocity (Fig 2D-2F). The PSE was determined by fitting a normal (Gaussian) cumulative distribution function using the fit function in Matlab (version R2019b) for data collected from otherwise similar staircases that started in opposite directions (Fig. 2). This determined the mean of the psychometric function (also the PSE) at which responses were equally likely to be reported in each direction. The slope of the psychometric function (i.e., sigma or standard deviation) was also determined. In the current study, sigma denotes the slope of the psychometric function in an individual while standard deviation will be used to describe the variation around the mean in a population. In the example shown, offsetting the normal visual stimulus 60° to the left (Fig 2A) tended to bias the

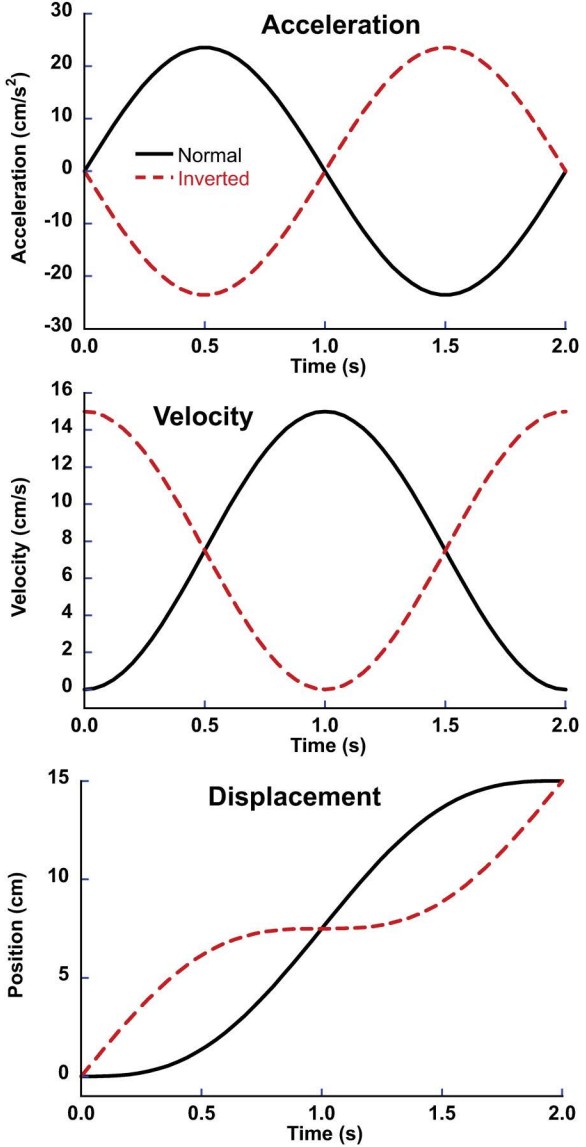

**Fig 1. Each trial block included stimuli in which the visual motion profile matched the inertial motion (normal, solid line) such that the visual stimulus was consistent with moving through a fixed environment.** It also included stimuli in which the velocity profile of the visual stimulus was inverted such that it started and ended at the peak velocity (15 cm/s) while slowing to zero in the middle (inverted, dashed line). Thus, the inverted stimulus was inconsistent with the inertial motion experience.

perception of inertial direction such that an inertial stimulus had to be shifted 10.3° to the right to be perceived as neutral (i.e., straight ahead). The opposite bias was seen when the visual stimulus was shifted 60° right (Fig 2C). Visual offsets produced smaller biases with the inverted velocity stimulus (Fig 2D and 2F). In both cases, the zero-offsets produced very small biases (Fig 2B and 2E).

Statistics were performed with JMP for the Macintosh (version 18.2.0). A two-way analysis of variance (ANOVA) was done using the three non-zero offsets (45, 60, and 75°) and the two visual stimulus types (normal and inverted). A one-way ANOVA followed by a post-hoc analysis were performed for analysis of heading offsets and effects among participants. A Student's paired T-test with alpha specified at 0.05 for the two types of visual stimulus profile.

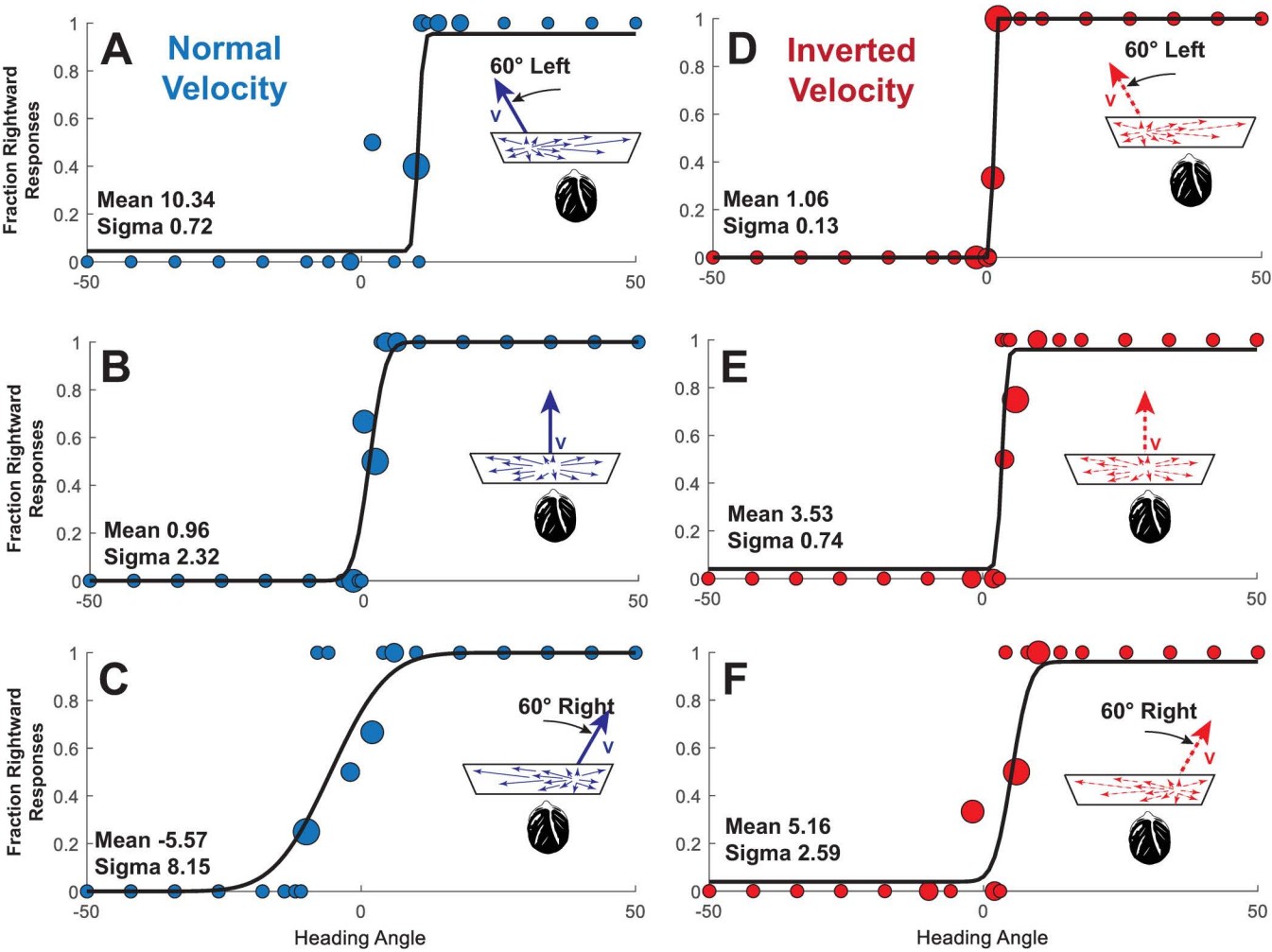

**Fig 2. Example data from an individual subject (#18) tested in a trial block with a ±60° offset.** With normal velocity visual stimuli (panels A-C), the visual stimulus influenced the perceived inertial heading. Thus, for a visual stimulus shifted to the left. a straightforward inertial heading (0°) is more likely to be perceived as left, which corresponds to the cumulative distribution function being shifted to the right (Panel A). For this subject, the inverted stimulus (panels D-F) had a minimal effect on inertial heading perception. Each small circle represents an individual stimulus presentation, larger circles represent multiple stimulus presentations in proportion to their diameter.

## Results

Heading perception was biased in the direction of the offset visual stimulus and the mean (bias) and sigma were determined from individual responses (Fig 2). When the aggregate data was examined, the inverted visual stimulus biased inertial perception less than the velocity matched stimulus (Fig 3). When the stimuli in which the visual stimulus was offset (i.e., aligned stimuli were excluded) the mean bias towards the visual stimulus was 6.0 ± 1.0° (mean ± SE) with inverted visual stimulus and 9.4 ± 1.2° (mean ± SD) with a normal visual stimulus. A three-way ANOVA was performed using visual stimulus type (normal, inverted) and the three non-zero offsets (45, 60, 75°). This (ANOVA, $F_{(1,38)}$ = 9.61) revealed that visual stimulus type is the only statistically significant factor (p = 0.0022) with no effect of offset magnitude ($F_{(2, 38)}$ = 0.28, p = 0.76). There was a significant effect between subjects (ANOVA, $F_{(6,19)}$ = 13.2, p < 0.0001) A paired T-test of inverted vs. normal visual stimulus was statistically significant t(118) = 6.487, p < 0.0001 (two-tailed) when all the non-zero offsets

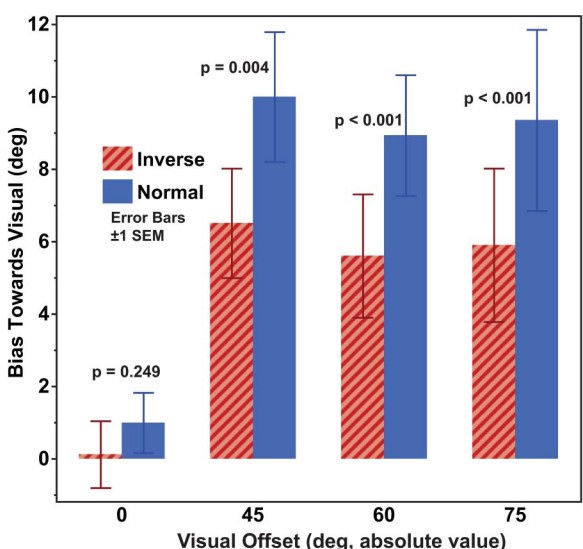

**Fig 3. Effect of visual offset for inverted (striped) and normal (solid) visual stimuli.** The data shown represent all twenty subjects. For no visual offset (zero) the bias is shown as towards the right. When the visual stimulus was offset, the bias was shown towards the visual stimulus so that leftward and rightward biases could be shown together. Error bars represent ±1 SEM. P-values are calculated using a paired T-test and are shown for each offset.

were considered. When the offsets were considered separately, the effect remained statistically significant at 45° t(39) = 3.04, p = 0.004, 60° t(39) = 4.18, p = 0.0002, and 75° t(39) = 4.58, p < 0.0001. Given that the size of the offset did not have a significant effect, the non-zero offsets were combined to show the individual data so that the effect of the visual stimulus could be seen for each of the twenty subjects (Fig 4).

For 17 of the 20 participants, the acceleration-matched visual stimulus biased perceived heading direction more than an inverted velocity stimulus (Fig 4). However, the amount of bias varied between subjects, as did the amounts of relative bias. This variation between subjects was statistically significant (ANOVA, F = 4.4, df = 19, p < 0.0001). There was one subject (#17) with an unusually large bias, while in two subjects (#1 and #8), the visual stimulus of either type had effectively no influence on inertial heading direction. Two subjects had a bias with the normal visual stimulus but not with the inverted one (#2, 20). In an additional 4 subjects, the bias induced by the normal stimulus was at least twice that of the inverse visual stimulus (#3, 5, 10, 18). In most of the remaining subjects, there was a modest bias towards the visual stimulus, which was greater with a normal visual stimulus than an inverted one.

The slope of the psychometric function (sigma) was also determined (Fig 5). When the potential effects of visual stimulus type and offset were examined there were no statistically significant effects using a three-way ANOVA to look at stimulus type and non-zero offsets. When the visual stimulus profile was examined (ANOVA, F(1,38) = 0.36) revealed that the stimulus type (p = 0.55) was not a significant factor. When offset magnitude was examined (ANOVA. F(2,38) = 0.20) there was also no significant effect (p = 0.82). As with the bias, there was significant variation in sigma between subjects (ANOVA, F(6,19) = 3.38, p < 0.0001). Overall, the normal stimulus had a sigma of 6.1 ± 0.4 (SEM) and the inverted stimulus had a sigma of 5.6 ± 0.4 (SEM), but this difference was not statistically significant t(179) = -0.94, p = 0.35. There was also no effect of the size of the visual offset (ANOVA, F = 0.16, df = 3, p = 0.92). Sigma was not correlated with the magnitude of bias towards the visual stimulus (R$^2$ = 0.009, p = 0.07).

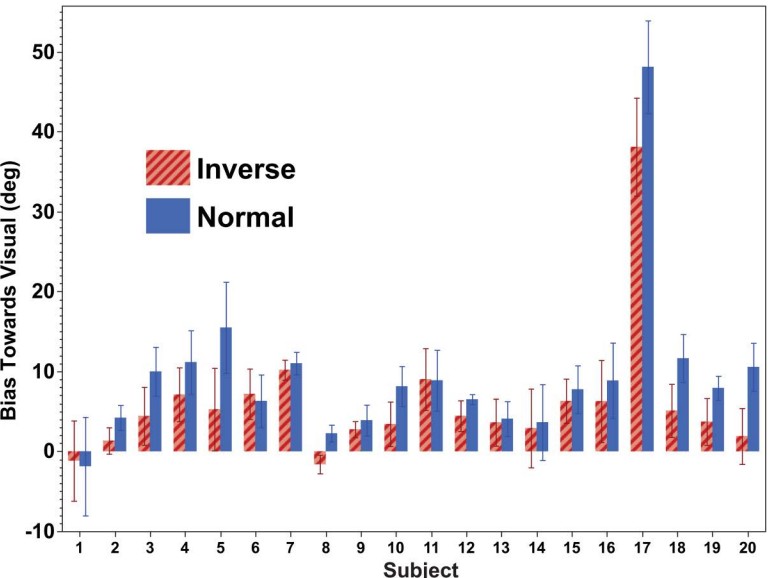

**Fig 4. Effect of visual offset for inverted (striped) and normal (solid) visual stimuli by subject.** Only non-zero visual offsets are included. Bias is plotted towards the visual stimulus offset and represents both leftward and rightward biases. Error bars represent ±1 SEM, which was calculated across all visual offsets (±45, ±60, and ±75°) collected in that subject.

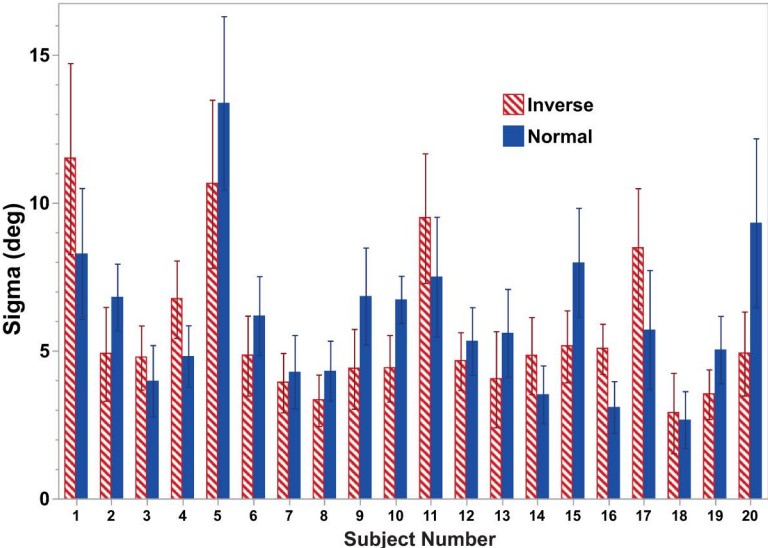

**Fig 5. Sigma for inverted (striped) and normal (solid) visual stimuli by subject.** All visual offsets were included (including zero) as there was no significant difference based on offset size. Error bars represent ±1 SEM.

## Discussion

It was previously reported that visual and inertial heading stimuli were optimally integrated even when the visual stimulus did not have a motion profile consistent with the inertial motion [11]. In that study, the visual stimulus was inconsistent with inertial motion because of a constant velocity, and beyond that, the type of inconsistency was not explored. This finding

was somewhat surprising to the current authors as other forms of inconsistency, such as large directional offsets between stimuli [5] and timing differences beyond about 200 ms [12] diminish visual-inertial multisensory integration. Furthermore, when visual stimuli are presented alone, they have a more compelling sense of vection when there is an acceleration component and are not just a fixed velocity [13], suggesting an accelerating visual stimulus is more likely to be perceived as self-motion. Unpublished data from our group confirmed the findings of Butler et al, using a 2s stimulus, suggesting the finding was not an artifact of the 1s stimulus used or other parameters that may have been specific to the prior study.

During the common situation of motion through a static environment, visual and inertia cues have a tight causal relationship that allows sensory integration to occur [1,25–27]. However not all visual motion corresponds to self-motion, for example, walking through blowing snow or moving with a crowd. Visual motion alone should not be interpreted as self-motion, but there are situations when this occurs including the false perception of self-motion induced by visual motion or vection [13,28–32] and visually induced motion sickness [33,34].

When common causation between two sensory modalities is not plausible, they should not be integrated. In the current study, even though the visual stimulus isn't plausible as occurring due to inertial motion, there was still integration but the influence of the incongruent visual stimulus was diminished to. on average, 64% of the congruent one. There were two subjects (#2, 20, Fig 4) in whom the normal visual stimulus biased inertial perception, and the inverted visual stimulus did not, so in most subjects, even the inverted visual stimulus biased inertial perception. Interestingly, subjects 2 and 20, aged 31 and 29 respectively, are the oldest participants in the study. This raises the possibility of considering the impact of age on inertial heading perception in future research.

There was significant variation between subjects. Subject #17 was an outlier in that their bias towards visual stimulus was much larger than any other subjects (Fig 4). An argument could be made for excluding this subject as a statistical outlier, but we chose to leave him in. The subject was an 18-year-old white male, so he was not an outlier in terms of demographics. The sigma for this subject (Fig 5) was not an outlier. One possible interpretation of this subject's response was that they did not rigorously follow the instructions and reported the direction of the visual stimulus instead of the inertial stimulus. However, this doesn't fully explain the findings as the bias was still less than the offset, and the normal visual stimulus still had a larger bias than the inverted one.

Previous work in our lab has looked at the temporal alignment of visual and inertial stimuli [12] and found that for misalignments of 500 ms or longer, the visual heading direction no longer biases inertial heading perception. One way to think about the current study would be that the visual and inertial stimuli are 180° out of phase, or since a 2s stimulus was used, they are temporally shifted by 1s. However, this is different than the type of shifts used previously [12] since the visual and inertial stimuli begin and end at the same time. The fact that visual-inertial integration still occurs despite a 1s temporal offset in peak velocity and peak acceleration (Fig 1) implies factors other than when the peak velocity or peak acceleration occurs to determine the effect described with a timing delay. A future direction could be to independently vary the relative start and end point of these stimuli to gain further insight into what factors are important for this integration to occur.

The current experiments did not include any unisensory conditions (I,e. visual only or inertial only) although such experiments have been previously published in our laboratory [4,20]. We did not want to include a visual only condition because subjects were specifically asked to identify the inertial heading, and it would likely be confusing. Also, the reliability of the visual stimulus has shown to be strongly direction dependent with decreased reliability with more lateral headings [35] which would make a meaningful unisensory visual condition difficult to design and interpret. Although an inertial only condition could have quantified underlying biases, these were ultimately cancelled by averaging the effects of visual offsets in opposite directions.

It is well known that multisensory integration relies on the relative reliability of the stimuli [1,18,36–39]. The current findings could be potentially explained if the inverted visual stimulus was perceived as less reliable than the normal visual stimulus. Butler et al. found the raised cosine was more reliable than the constant velocity [11] although both were robustly integrated. In this study, the difference in the slope (sigma) of the psychometric function fits between the two types of

stimuli was not statistically significant (paired t-test, p = 0.4). This suggests that the perceived reliability of the stimuli did not explain the observed larger bias in the normal visual stimulus.

In the current study, there were no attempts to control or measure gaze position. Gaze position was controlled and measured in previous studies from our group [4,20,35]. This was not done in the current study because it is difficult to maintain a gaze position without a fixation point, and when a fixation point is present subjects tend to judge the heading by reporting the focus expansion relative to the fixation point. In the current study, subjects were instructed to report only the direction of the inertial stimulus, which has previously been demonstrated to be independent of eye position [19,35]. Thus, the effect of gaze position in the current paradigm is unknown but likely minimal.

The size of the visual offset in this study didn't make a difference in either the size of the bias or the sigma of the psychometric function fit. Multiple offsets were included because one hypothesis was that common causation might be perceived at a larger offset if the visual stimulus was consistent with the acceleration profile, but this was not borne out in the data. The smallest offset tested was 45° which was large compared with the bias, and this may be why the bias did not depend on the size of the offset. The bias may reach some maximal value, after which further visual offsets have no effect. We have also found that the reliability of the visual stimulus decreases as it gets more laterally displaced [35] so that larger displacements may have been counteracted by a lower perceived reliability. It is possible, if not likely, that smaller offsets would have produced a smaller bias, but this is outside of the scope of the current experiments.

It may be advantageous to integrate visual and inertial signals when they are not in agreement. Once linear motion reaches a constant velocity there is no acceleration and beyond vibration that may occur as a result there is motion without a corresponding inertial perception. Because this only occurs at a constant velocity, it may explain the earlier finding that a constant velocity visual stimulus has the same influence as a velocity match stimulus [11]. The current study demonstrates that visual-inertial integration is negatively impacted once the visual stimulus includes acceleration components that are incongruent with those of the inertial stimulus. This effect was statistically significant (p < 0.0001), thus, invalidating the last two hypotheses predicting a small or independent influence of visual motion profile.

## Supporting information

**S1 File. VisAccel complete data_redacted2.xlsx**
(XLSX)

## Acknowledgments

The authors would like to thank Cesar Arduino for providing technical assistance for the experiments.

## Author contributions

**Conceptualization:** Eric Anson, Benjamin T. Crane.

**Data curation:** Benjamin T. Crane.

**Formal analysis:** Miguel A. Yakouma, Benjamin T. Crane.

**Funding acquisition:** Benjamin T. Crane.

**Investigation:** Miguel A. Yakouma, Benjamin T. Crane.

**Methodology:** Miguel A. Yakouma, Eric Anson, Benjamin T. Crane.

**Project administration:** Benjamin T. Crane.

**Resources:** Benjamin T. Crane.

**Software:** Benjamin T. Crane.

**Supervision:** Benjamin T. Crane.

**Visualization:** Benjamin T. Crane.

**Writing – original draft:** Benjamin T. Crane.

**Writing – review & editing:** Miguel A. Yakouma, Eric Anson, Benjamin T. Crane.

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
