## [Decision Letter · Decision Letter 0]

31 Jul 2024

PONE-D-24-23927Effect of Inverted Visual Acceleration Profile on Vestibular Heading Perception.PLOS ONE

Dear Dr. Crane,

Thank you for submitting your manuscript to PLOS ONE. After careful consideration, we feel that it has merit but does not fully meet PLOS ONE’s publication criteria as it currently stands. Therefore, we invite you to submit a revised version of the manuscript that addresses the points raised during the review process.

As you will see below, while the expert reviewers were mainly positive about your work, they also raised some concerns, most notably the large between-subject variability (Reviews #1 and #2), the lack of unimodal baseline conditions and eye-tracking recordings (Review #3), and potential differences in the reliability of the responses between the two visual motion profiles (Review #4). Please address these concerns in your revision of the manuscript. I will likely send out the new version to some or all of the previous reviewers.

We look forward to receiving your revised manuscript.

Kind regards,

Patrick Bruns

Academic Editor

PLOS ONE

Journal Requirements:

2. Thank you for stating the following financial disclosure: "Grant support was provided by 2 R01 DC013580. EA was supported in part by the National Institutes of Health (NIDCD K23 DC018303)"  

3. Thank you for stating the following in the Acknowledgments Section of your manuscript: "The authors would like to thank Cesar Arduino for providing technical assistance for the experiments. Grant support was provided by 2 R01 DC013580. EA was supported in part by the National Institutes of Health (NIDCD K23 DC018303).

Manuscript Click here to access/download;Manuscript;BTC Vis Accel PLoS

v2.docx Yakouma, et al. Inverted Visual and Vestibular Heading Perception"

Please remove any funding-related text from the manuscript and let us know how you would like to update your Funding Statement. Currently, your Funding Statement reads as follows: "Grant support was provided by 2 R01 DC013580. EA was supported in part by the National Institutes of Health (NIDCD K23 DC018303)"

Reviewers' comments:

Reviewer's Responses to Questions

**Comments to the Author**

1. Is the manuscript technically sound, and do the data support the conclusions?

Reviewer #1: Yes

Reviewer #2: Yes

Reviewer #3: Yes

Reviewer #4: Partly

2. Has the statistical analysis been performed appropriately and rigorously? 

Reviewer #1: Yes

Reviewer #2: Yes

Reviewer #3: Yes

Reviewer #4: No

3. Have the authors made all data underlying the findings in their manuscript fully available?

Reviewer #1: Yes

Reviewer #2: Yes

Reviewer #3: Yes

Reviewer #4: No

4. Is the manuscript presented in an intelligible fashion and written in standard English?

Reviewer #1: Yes

Reviewer #2: Yes

Reviewer #3: Yes

Reviewer #4: Yes

5. Review Comments to the Author

Reviewer #1: In the current study, the author presented visual stimuli with different temporal profile, namely, inverted, compared to vestibular, and found that it generated less impact on the the combined performance. The results are clear. The manuscript is well written. I do not have many comments. The only query is that although the results are quite clear for the mean PSE shift comparison between inverted and normal visual stimulation condition, yet they are quite varied for individual subjects (figure 4).

Reviewer #2: This study investigates the integration of visual and inertial cues in influencing heading perception during motion. The authors found that inverted visual stimuli consistently biased inertial heading perception less than matched velocity stimuli across different offsets. This represents a novel and valuable contribution to the field of sensory integration and perception. The manuscript is generally well-organized, but I have a few specific comments listed below:

Abstract:

There is a spelling mistake for “whish,” which should be corrected.

Introduction:

It would be beneficial to provide more explanation for the choice of large offset values (±45°, ±60°, ±75°). Subjects might use the discrepancy between visual and vestibular cues to infer their causal relationship (refer to Acerbi L, Dokka K, Angelaki DE, Ma WJ, 2018. Bayesian comparison of explicit and implicit causal inference strategies in multisensory heading perception. PLoS Comput Biol 14(7): e1006110).

Numbers and units are sometimes separated by spaces, and sometimes they are not, which creates inconsistency, for example, '2s' and '2 s'.

Results: More detailed descriptions of the figures within the text are needed, explaining what each figure shows and its significance.

The variation between subjects is noted but not explored in depth. Potential reasons for individual differences in bias should be discussed, including how these differences might affect the overall results.

Discussion:

The interpretation of the results is somewhat superficial. A deeper analysis of the implications of the findings and their advancement of the field is needed.

Provide a more in-depth interpretation of the results, considering alternative explanations and the broader implications.

Bibliography:

Several instances of “Deangelis” are misspelled and should be corrected to “DeAngelis.”

Reviewer #3: The authors present the results of a psychophysical study on heading perception of n=20 healthy participants while they experienced 2 s episodes of passive inertial self motion (15 cm displacements on a 6DoF motion platform, see Fig. 1). Participants viewed random-dot kinematic (RDK) noise fields that simulated congruent (in phase with platform motion) or incongruent (out-of-phase with platform motion) optic flow sequences. The focus of expansion (FoE) of the flow fields was offset +/- 45, 60, or 75 degrees. The hypotheses test the effects of phase differences between visual (RDKs) and vestibular (self motion on platform) and direction offsets on the subjective point of straight ahead self motion. The results indicate that the direction offset of the FoE of RDK visual motion biases estimates of straight ahead self motion in participants who viewed in-phase (congruent) visual RDK flow fields more compared to conditions where the velocity profile of the RDK flow fields was out of phase to the platform motion (Fig. 3). The results are discussed in terms of multisensory integration of visual and vestibular cues of self motion.

The work appears to have been carefully conducted. Some clarifications are needed (see below).

Major revision requests

1) It seems that some control conditions could have been included to determine the extent that the participants were truly integrating visual and vestibular cues of self motion. The most obvious ones would be to place the participants blindfolded on the motion platform to determine their acuity of sensing "straight ahead" self motion without visual input. In the same manner, the participants could have been asked to judge the FoE of RDK flow fields (left or right of straight ahead) in the complete absence of platform motion. The results of these control conditions would have provided a "baseline" for direction judgments about these visual and vestibular stimuli.

2) There is no mention of eye-movement recordings during the different visual-vestibular stimulus conditions. The stimulus displays depicted in Fig. 2 show no indication of any stationary fixation marks in the RDK flow fields. Were the participants instructed to direct their gaze "straight ahead" and if so how was compliance monitored during trials with platform motions? Participants would naturally direct their gaze to the focus of expansion in flow fields. Did the authors note such behaviour and if so how did it influence their judgments about straight ahead with respect to their own self motion?

3) The data from participant #17 should be removed as a statistical outlier and the data analysis should be performed without these outlier data. It is unclear why P#17 was perceiving inertial self motion at +/-45 degrees as "straight ahead" more or less independent of the visual motion cues.

Minor requests for revision

4) Although the writing and English usage is in most cases standard, there is considerable room for improvement. Many small typos (e.g., in the Abstract "whish" should be "which") are evident throughout. Also English usage needs to be checked:

p. 3: "they must share self-motion as the common causation" should be "they must be consistent with self-motion as the

common causation".

p. 3: "Perception of heading direction is an ecologically relevant situation..." should be something like "Perception of

heading direction is an ecologically relevant sensory-based ability..."

p. 3: "... been shown in the current laboratory:" should be "... been shown by our group:"

p. 3: "... (i.e. the perception of self-motion induced by a visual stimulus while stationary) should be "... (i.e. the illusory

perception of self-motion induced by a visual stimulus while the observer is stationary)

p. 4: "... and inertial motion seems required for ..." should be "... and inertial motion seems to be required for ..."

p. 5: "... disparity angle of visual ..." should be "... angular deviation of visual ..."

p. 7: "The subjects wore a helmet that was coupled to the motion platform while ..." The use of the verb "coupled" needs

to be explained here. Was head motion measured independently of platform motion and if so how?

p.9: "... re-presented" should be "presented again ..."

p. 9: "... cumulative distribution function". Unclear if this distribution was normal (Gaussian) or otherwise.

p. 11: "... p. 3: "... in the current laboratory:" should be "... from our group:"

p. 11: "walking through blowing slow ..." should be "walking through blowing snow ..."

p. 11: "... but the influence of the inverted visual stimulus..." could be "... but the influence of the incongruent visual

stimulus..."

p. 11: "... so in most subjects the inverted visual stimulus biased inertial perception." could be "... so in most subjects even the inverted visual stimulus biased inertial perception."

p. 11: "average 64% of the normal one." could be "average 64% of the congruent one."

p. 12: "components that don’t match the visual stimulus." should be "components that are incongruent with those of the visual stimulus."

Reviewer #4: This paper investigates the role of visual motion profile on visual vestibular integration. The authors use two motion profiles one that is consistent with the velocity of the vestibular motion profile and one that is the inverted form of the profile. The paper is clearly written, and it is always nice to see the individual participant data.

I have some comments about the interpretation and analysis.

Comments:

Introduction/Discussion

The data shows that both motion profiles are integrated with the vestibular information as illustrated by the shift in the visual-vestibular PSE. Furthermore, the shift is less impacted by the inverted motion profile than the consistent motion profile. The authors posit that this is I'm the inverted motion profile is less it is integrated. While it is not unreasonable, another possibility is that there is a difference in the reliability of the responses between the two visual motion profiles, which was the case in Butler et al 2014, the raised cosine was more reliable than the constant velocity. This would mean that the inverted motion profile would be weighted less due to reliability and not causation. The authors could address this by analyzing the standard deviation (sigma) values of the fits of the cumulative distribution functions. If this cannot be done the authors should mention this possible caveat in the discussion.

It looks like there is no effect of visual offset (see below for suggested analysis). Were the authors surprised by this and could they write a line about it in the discussion.

Analysis

The data were fit to a cumulative distribution function. If this function is Gaussian, please explicitly state this.

The statistical analysis seems to be conducted in the wrong order; the ANOVA should be performed first, followed by t-tests as post hoc analyses. The appropriate ANOVA should be a 2 Motion Profile (Normal, Inverted) × 3 Offset (30, 45, 60) ANOVA. While it is mentioned that this analysis was conducted, the results concerning the offset are not discussed in the results section. Regardless of whether the results are significant, they are of interest and should be addressed. This analysis would allow for a discussion on both profile and offset. If feasible, this ANOVA should be conducted on both the PSE and the sigma of the fits.

Results

To enhance clarity, it would be helpful to provide additional details about Figure 2 at the beginning of the Results section to better link the mean, bias, and PSE.

In the second paragraph of the Results section, the ANOVA is mentioned. Could the authors clarify this point? Additionally, more information on the individual participant analysis would be appreciated, particularly regarding how the standard error of the mean (SEM) was calculated.

Very minor comment:

there's a typo in the abstract; whish should be which.

Finally, here are some papers that might be of interest to the authors that seemed relevant to this work

Butler, J. S., Smith, S. T., Campos, J. L., & Bülthoff, H. H. (2010). Bayesian integration of visual and vestibular signals for heading. Journal of vision, 10(11), 23-23.

Butler, J. S., Campos, J. L., Bülthoff, H. H., & Smith, S. T. (2011). The role of stereo vision in visual–vestibular integration. Seeing and perceiving, 24(5), 453-470.

Drugowitsch, J., DeAngelis, G. C., Klier, E. M., Angelaki, D. E., & Pouget, A. (2014). Optimal multisensory decision-making in a reaction-time task. Elife, 3, e03005.

Fetsch, C. R., Pouget, A., DeAngelis, G. C., & Angelaki, D. E. (2012). Neural correlates of reliability-based cue weighting during multisensory integration. Nature neuroscience, 15(1), 146-154.

Rideaux, R., Storrs, K. R., Maiello, G., & Welchman, A. E. (2021). How multisensory neurons solve causal inference. Proceedings of the National Academy of Sciences, 118(32), e2106235118.

Yakubovich, S., Israeli-Korn, S., Halperin, O., Yahalom, G., Hassin-Baer, S., & Zaidel, A. (2020). Visual self-motion cues are impaired yet overweighted during visual–vestibular integration in Parkinson’s disease. Brain communications, 2(1), fcaa035.

Zaidel, A., Goin-Kochel, R. P., & Angelaki, D. E. (2015). Self-motion perception in autism is compromised by visual noise but integrated optimally across multiple senses. Proceedings of the National Academy of Sciences, 112(20), 6461-6466.

Zhou, L., & Gu, Y. (2023). Cortical mechanisms of multisensory linear self-motion perception. Neuroscience Bulletin, 39(1), 125-137.

6. PLOS authors have the option to publish the peer review history of their article (what does this mean? ). If published, this will include your full peer review and any attached files.

**Do you want your identity to be public for this peer review?** For information about this choice, including consent withdrawal, please see our Privacy Policy .

Reviewer #1: No

Reviewer #2: No

Reviewer #3: No

Reviewer #4: No

---

## [Author Response · Author response to Decision Letter 1]

21 Feb 2025

See the attached document. Also this has been pasted here:

However, funding information should not appear in the Acknowledgments section or other areas of your manuscript. We will only publish funding information present in the Funding Statement section of the online submission form.

Please remove any funding-related text from the manuscript and let us know how you would like to update your Funding Statement. Currently, your Funding Statement reads as follows: "Grant support was provided by 2 R01 DC013580. EA was supported in part by the National Institutes of Health (NIDCD K23 DC018303)"

The funding was removed from the acknowledgments. The funding statement has been updated.

Reviewers' comments:

Reviewer's Responses to Questions

Comments to the Author

1. Is the manuscript technically sound, and do the data support the conclusions?

Reviewer #1: Yes

Reviewer #2: Yes

Reviewer #3: Yes

Reviewer #4: Partly

2. Has the statistical analysis been performed appropriately and rigorously?

Reviewer #1: Yes

Reviewer #2: Yes

Reviewer #3: Yes

Reviewer #4: No

3. Have the authors made all data underlying the findings in their manuscript fully available?

Reviewer #1: Yes

Reviewer #2: Yes

Reviewer #3: Yes

Reviewer #4: No

4. Is the manuscript presented in an intelligible fashion and written in standard English?

Reviewer #1: Yes

Reviewer #2: Yes

Reviewer #3: Yes

Reviewer #4: Yes

5. Review Comments to the Author

Reviewer #1: In the current study, the author presented visual stimuli with different temporal profile, namely, inverted, compared to vestibular, and found that it generated less impact on the the combined performance. The results are clear. The manuscript is well written. I do not have many comments. The only query is that although the results are quite clear for the mean PSE shift comparison between inverted and normal visual stimulation condition, yet they are quite varied for individual subjects (figure 4).

Thanks. We agree that there was a lot of individual variation. In our experience, this is fairly typical of these types of human psychophysics experiments.

Reviewer #2: This study investigates the integration of visual and inertial cues in influencing heading perception during motion. The authors found that inverted visual stimuli consistently biased inertial heading perception less than matched velocity stimuli across different offsets. This represents a novel and valuable contribution to the field of sensory integration and perception. The manuscript is generally well-organized, but I have a few specific comments listed below:

Thanks. We agree.

Abstract:

There is a spelling mistake for “whish,” which should be corrected.

This was corrected to “which”

Introduction:

It would be beneficial to provide more explanation for the choice of large offset values (±45°, ±60°, ±75°). Subjects might use the discrepancy between visual and vestibular cues to infer their causal relationship (refer to Acerbi L, Dokka K, Angelaki DE, Ma WJ, 2018. Bayesian comparison of explicit and implicit causal inference strategies in multisensory heading perception. PLoS Comput Biol 14(7): e1006110).

Thanks. The Acerbi et al paper used smaller offsets (5, 10, 20, and 40°). The offsets we used were based on previous studies in which visual stimuli could bias the vestibular heading direction with offsets up to 90° [Rodriguez, 2019 #2059] even though beyond about 30° subjects were aware these stimuli did not have a causal relationship [Rodriguez, 2020 #2084]. In the current study, we weren’t interested in determining if subjects inferred a causal relationship because the lack of a cause relationship would have been obvious from the inverted velocity profile of the visual stimulus. We now make this clear in the introduction:

“The purpose of this study was not to look at causal inference[18] as it would have been obvious to most observers that the inverted visual stimulus was inconsistent with the inertial motion. We had previously observed that visual stimuli bias inertial direction perception even with large offsets of up to 90° when it was clear to subjects that the stimuli were artificially offset[5]. Thus, the range of offsets used in the current study went to 75°.”

Numbers and units are sometimes separated by spaces, and sometimes they are not, which creates inconsistency, for example, '2s' and '2 s'.

Thanks for noticing this. We have now standardized to not have a space between the number and the s (e.g. 2s).

Results: More detailed descriptions of the figures within the text are needed, explaining what each figure shows and its significance.

Thanks. We now explain in detail each panel for Figure 1:

“The inverted visual stimulus also lasted 2s (Fig. 1) and had a similar peak acceleration (Fig. 1A), peak velocity (Fig. 1B), and displacement (Fig. 1C. The difference between the normal and inverted visual stimulus was that the inverted stimulus began and ended at the peak velocity while the normal stimulus achieved the peak velocity at the midpoint (Fig. 1B). However, both stimuli were consistent with the same displacement, and had the same peak velocity and peak acceleration.”

For figure 2, the details of the figure are explained in more detail in the Analysis section:

“The fraction of rightward responses for each stimulus level was plotted as a function of the heading direction tested for both the normal velocity (Fig. 2A-C) and inverted velocity (Fig. 2 D-F). The PSE was determined by fitting a cumulative distribution function using the fit function in Matlab (version R2019b) for data collected from otherwise similar staircases that started in opposite directions (Fig. 2). This determined the mean of the psychometric function (also the PSE) at which responses were equally likely to be reported in each direction. The width of the psychometric function (i.e. sigma or standard deviation of the psychometric function) was also determined. In the example shown, offsetting the normal visual stimulus to the left (Fig. 2A) tended to bias the perception of inertial direction such that an inertial stimulus had to be shifted 10.3° to the right to be perceived as neutral (i.e. straight ahead). The opposite bias was seen when the visual stimulus was shifted towards the right (Fig. 2C). Visual offsets produced smaller biases with the inverted velocity stimulus (Figs. 2D and 2F).”

Figure 3 is a simple figure that is explained at the start of the Results. It is now qualified that this shows the aggregate data.

Figure 4 is now described in greater detail in the text. At the end of the first paragraph of the results it now says: “Given that the size of the offset did not have a significant effect, the non-zero offsets were combined to show the individual data so that the effect of the visual stimulus could be seen for each of the twenty subjects (Fig. 4).”

The variation between subjects is noted but not explored in depth. Potential reasons for individual differences in bias should be discussed, including how these differences might affect the overall results.

Other reviewers raised similar criticisms. The variation is now better described in the results as well as the discussion.

Discussion:

The interpretation of the results is somewhat superficial. A deeper analysis of the implications of the findings and their advancement of the field is needed.

Provide a more in-depth interpretation of the results, considering alternative explanations and the broader implications.

Thanks, similar comments were also made by other reviewers, and this is now included. We now discuss the variation between subjects and have also presented and discussed the width of the psychometric fits (sigma). We also now discuss the possibility that the reliability of the normal and inverted visual stimuli may have caused them to be perceived differently. The observation that the bias did not depend on the size of the offset is also now discussed.

Bibliography:

Several instances of “Deangelis” are misspelled and should be corrected to “DeAngelis.”

Thanks for noticing this. It has been corrected to “DeAngelis”

Reviewer #3: The authors present the results of a psychophysical study on heading perception of n=20 healthy participants while they experienced 2 s episodes of passive inertial self motion (15 cm displacements on a 6DoF motion platform, see Fig. 1). Participants viewed random-dot kinematic (RDK) noise fields that simulated congruent (in phase with platform motion) or incongruent (out-of-phase with platform motion) optic flow sequences. The focus of expansion (FoE) of the flow fields was offset +/- 45, 60, or 75 degrees. The hypotheses test the effects of phase differences between visual (RDKs) and vestibular (self motion on platform) and direction offsets on the subjective point of straight ahead self motion. The results indicate that the direction offset of the FoE of RDK visual motion biases estimates of straight ahead self motion in participants who viewed in-phase (congruent) visual RDK flow fields more compared to conditions where the velocity profile of the RDK flow fields was out of phase to the platform motion (Fig. 3). The results are discussed in terms of multisensory integration of visual and vestibular cues of self motion.

The work appears to have been carefully conducted. Some clarifications are needed (see below).

Thanks

Major revision requests

1) It seems that some control conditions could have been included to determine the extent that the participants were truly integrating visual and vestibular cues of self motion. The most obvious ones would be to place the participants blindfolded on the motion platform to determine their acuity of sensing "straight ahead" self motion without visual input. In the same manner, the participants could have been asked to judge the FoE of RDK flow fields (left or right of straight ahead) in the complete absence of platform motion. The results of these control conditions would have provided a "baseline" for direction judgments about these visual and vestibular stimuli.

We have done and published these types of control experiments previously, although in a different subject panel as a part of unrelated experiments[Crane, 2017 #1802][Rodriguez, 2018 #1964][Crane, 2012 #1280]. What previous work on this has shown is that subjects are able to judge visual headings with a high degree of accuracy and precision. Inertial headings in darkness were less precise and also would often have a small bias.

In the current study, subjects were asked to judge only the inertial direction. We felt it would be confusing to include conditions where no inertial stimulus was present and visual headings were reported as this might encourage subjects to report the visual heading in subsequent experiments. Furthermore, the reliability of the visual stimulus has been shown to decrease as the heading gets further from straight ahead. Obtaining unisensory control data for the range of visual headings included would have involved another series of experiments that would have greatly complicated the data collection and wasn’t done.

In the current experiments, the control condition was when the visual heading was not offset, and this provides the “baseline” that the reviewer requests. We understand that the performance of an inertial heading in darkness may be different than when a visual heading was available.

At this point, we no longer have access to many of the subjects in this panel as it is not possible to do additional control experiments.

The following has been added to the discussion:

“The current experiments did not include any unisensory conditions (I,e. visual only or initial only) although such experiments have been previously published in our laboratory[Crane, 2017 #1802][Rodriguez, 2018 #1964][Crane, 2017 #1802]. We did not want to include a visual-only condition because subjects were specifically asked to identify the inertial heading, and it would likely be confusing. Although an inertial-only condition could have quantified underlying biases, these were ultimately cancelled by averaging the effects of visual offsets in opposite directions.”

2) There is no mention of eye-movement recordings during the different visual-vestibular stimulus conditions. The stimulus displays depicted in Fig. 2 show no indication of any stationary fixation marks in the RDK flow fields. Were the participants instructed to direct their gaze "straight ahead" and if so how was compliance monitored during trials with platform motions? Participants would naturally direct their gaze to the focus of expansion in flow fields. Did the authors note such behaviour and if so how did it influence their judgments about straight ahead with respect to their own self motion?

Eye movements were not recorded in this study. No fixation point was provided, and participants were not given any instructions on where to look. We have previously published data where eye position was carefully monitored and controlled. Subjects were not given any instruction on gaze position. This is now clarified in the Methods: “No fixation point was provided, and subjects were not given any specific instructions on where to look.”

We have previously done experiments to examine the effect of gaze position [Rodriguez, 2018 #1964][Crane, 2012 #1280]. As the reviewer mentions it is difficult to maintain gaze at a particular location without a fixation point and when a fixation point is present, subjects tend to judge heading as the location of the focus of expansion relative to the fixation point. Subjects were asked to report only inertial heading in these experiments and this perception is in body coordinates and insensitive to eye position. Of course, it is possible that eye position had some effect. We added a paragraph to address this concern:

“In the current study, there were no attempts to control or measure gaze position. Gaze position was controlled and measured in some previous studies in the current laboratory[Rodriguez, 2018 #1964][Crane, 2012 #1280]. This was not done in the current study because it is difficult to maintain a gaze position without a fixation point and when a fixation point is present subjects tend to judge the heading by reporting the focus expansion relative to the fixation point. Subjects were instructed to report only the direction of the inertial stimulus which has previously been demonstrated to be independent of eye position[Crane, 2012 #1280][Cuturi, 2013 #1412]. Thus, the effect of gaze position i

---

## [Decision Letter · Decision Letter 1]

24 Mar 2025

PONE-D-24-23927R1Effect of Inverted Visual Acceleration Profile on Vestibular Heading Perception.PLOS ONE

Dear Dr. Crane,

Thank you for submitting your manuscript to PLOS ONE. After careful consideration, we feel that it has merit but does not fully meet PLOS ONE’s publication criteria as it currently stands. Therefore, we invite you to submit a revised version of the manuscript that addresses the points raised during the review process.

As you will see below, the reviewers were largely satisfied with your revisions. However, Reviewers #3 and #4 had a few remaining minor comments which should be addressed. In particular, I agree with Reviewer #4 that the ANOVA design should be clarified.

We look forward to receiving your revised manuscript.

Kind regards,

Patrick Bruns

Academic Editor

PLOS ONE

Journal Requirements:

Reviewers' comments:

Reviewer's Responses to Questions

**Comments to the Author**

1. If the authors have adequately addressed your comments raised in a previous round of review and you feel that this manuscript is now acceptable for publication, you may indicate that here to bypass the “Comments to the Author” section, enter your conflict of interest statement in the “Confidential to Editor” section, and submit your "Accept" recommendation.

Reviewer #2: All comments have been addressed

Reviewer #3: All comments have been addressed

Reviewer #4: All comments have been addressed

2. Is the manuscript technically sound, and do the data support the conclusions?

Reviewer #2: Yes

Reviewer #3: Yes

Reviewer #4: Yes

3. Has the statistical analysis been performed appropriately and rigorously? 

Reviewer #2: Yes

Reviewer #3: Yes

Reviewer #4: I Don't Know

4. Have the authors made all data underlying the findings in their manuscript fully available?

Reviewer #2: Yes

Reviewer #3: Yes

Reviewer #4: Yes

5. Is the manuscript presented in an intelligible fashion and written in standard English?

Reviewer #2: Yes

Reviewer #3: Yes

Reviewer #4: Yes

6. Review Comments to the Author

Reviewer #2: The authors have thoroughly addressed all of my questions, and I have no further comments at this time.

Reviewer #3: The authors have responded to each of my queries. However, the revised text needs some editing by a native speaker of English with experience in scientific writing.

Minor points requiring final revision

1) p. 10: "In the current manuscript ..." should state " In the current study ..."

2) p. 12 and elsewhere: "The width of the psychometric function ..." should state "The slope of the psychometric function ..."

3) p. 12 and elsewhere: "... significant effects (ANOVA, 5 DF, F-ratio 0.123)." should state "... significant effects (ANOVA, F = 0.123, df = 5)."

4) p. 12: "Sigma was poorly correlated ..." should state "Sigma was not correlated ..."

5) p. 14: "... we chose to leave it in." should either state "... we chose to leave him in." or rephrase the sentence as "An argument could be made for excluding data from this subject as statistical outliers but, we chose to leave them in."

6) p. 15: "...(I,e. visual only or initial only)..." should state "..(I,e. visual only or inertial only)..."

7) p. 16: "...in some previous studies in the current laboratory" should state ""...in previous studies from our group"

8) p. 17: "...This effect was statistically significant (p < 0.0001 ). Thus,..." should state "...This effect was statistically significant (p < 0.0001 ), thus..."

Reviewer #4: The authors have addressed most of my comments but I just want to clarify the ANOVA and how it has been reported.

It is a bit confusing as in the analysis the ANOVA is given as an offset x stimulus 3(45 deg, 60 deg and 70 deg)x2(normal and inverted).

But in the results it seems to be 6x1 (45 deg x normal, 60 deg x normal, 70 deg x normal, 45 deg x inverted, 60 deg x inverted and 70 deg x inverted).

If it is a 2x3 the DoF should be:

for the stimulus 1,19

for the offset 2,38

for the interaction 2,38

If it is a 6x1 then the DoF is 5 but it does not test the offset and stimulus separately, which means was a followup ANOVA conducted.

Can the authors clarify this.

What I would suggest is for the ANOVAs report the results as F(df1,df2)=?,p< and for the t-tests state t(df)=??,p<!--??.</p

7. PLOS authors have the option to publish the peer review history of their article (what does this mean? ). If published, this will include your full peer review and any attached files.

**Do you want your identity to be public for this peer review?** For information about this choice, including consent withdrawal, please see our Privacy Policy .

Reviewer #2: No

Reviewer #3: No

Reviewer #4: No

---

## [Author Response · Author response to Decision Letter 2]

1 Apr 2025

Reviewer #2: The authors have thoroughly addressed all of my questions, and I have no further comments at this time.

Response: Thanks

Reviewer #3: The authors have responded to each of my queries. However, the revised text needs some editing by a native speaker of English with experience in scientific writing.Minor points requiring final revision

1) p. 10: "In the current manuscript ..." should state " In the current study ..."

Response: This was fixed accordingly

2) p. 12 and elsewhere: "The width of the psychometric function ..." should state "The slope of the psychometric function ..."

Response: This was fixed accordingly

3) p. 12 and elsewhere: "... significant effects (ANOVA, 5 DF, F-ratio 0.123)." should state "... significant effects (ANOVA, F = 0.123, df = 5)."

Response: This was fixed accordingly

4) p. 12: "Sigma was poorly correlated ..." should state "Sigma was not correlated ..."

Response: This was fixed accordingly

5) p. 14: "... we chose to leave it in." should either state "... we chose to leave him in." or rephrase the sentence as "An argument could be made for excluding data from this subject as statistical outliers but, we chose to leave them in."

Response: This was fixed accordingly

6) p. 15: "...(I,e. visual only or initial only)..." should state "..(I,e. visual only or inertial only)..."

Response: This was fixed accordingly

7) p. 16: "...in some previous studies in the current laboratory" should state ""...in previous studies from our group"

Response: This was fixed accordingly

8) p. 17: "...This effect was statistically significant (p < 0.0001 ). Thus,..." should state "...This effect was statistically significant (p < 0.0001 ), thus..."

Response: This was fixed accordingly

Response: The authors thank the reviewer for what was clearly a careful reading of this revision. We have made the above changes as well as fixed other English grammar errors in the paper. The one exception was the initial two-way ANOVAs for mean and sigma that reviewer #4 had suggested changes to make more clear. Those are now presented in as a two-way ANOVA. In the previous version it was presented in a confusing way in which it was unclear if a one-way or two-way ANOVA was done.

Reviewer #4: The authors have addressed most of my comments but I just want to clarify the ANOVA and how it has been reported.It is a bit confusing as in the analysis the ANOVA is given as an offset x stimulus 3(45 deg, 60 deg and 70 deg)x2(normal and inverted).But in the results it seems to be 6x1 (45 deg x normal, 60 deg x normal, 70 deg x normal, 45 deg x inverted, 60 deg x inverted and 70 deg x inverted).If it is a 2x3 the DoF should be:for the stimulus 1,19for the offset 2,38for the interaction 2,38If it is a 6x1 then the DoF is 5 but it does not test the offset and stimulus separately, which means was a followup ANOVA conducted.Can the authors clarify this.What I would suggest is for the ANOVAs report the results as F(df1,df2)=?,p< and for the t-tests state t(df)=??,p

Response: Thanks for noticing this and pointing it out. The ANOVA was initially done as a two-way ANOVA as the reviewer suggests, although it was presented in a confusing way. Using the stimulus type (normal, inverted) and non-zero offsets (45, 60, and 70) we did a two-way ANOVA with df = 1, df = 2. The ANOVA was repeated as a two-way ANOVA.

This is now made clear in the Methods: ”A two-way analysis of variance (ANOVA) was done using the three non-zero offsets (45, 60, and 75°) and the two visual stimulus types (normal and inverted).”

It is also made clear in the Results with regard to the effect on mean: “A two-way ANOVA was performed using visual stimulus type (normal, inverted) and the three non-zero offsets (45, 60, 75°). This (ANOVA, F(1,2) = (4.88, 0.14)) revealed that visual stimulus type is the only statistically significant factor (p = 0.028) with no effect of offset magnitude (p = 0.87).”

A similar statement was changed for the effect on sigma: “When the potential effects of visual stimulus type and offset were examined there were no statistically significant effects using a two-way ANOVA to look at stimulus type and non-zero offsets (ANOVA, F(1,2) = (0.30, 0.17)) revealed that neither stimulus type (p = 0.58) nor offset magnitude (p = 0.84) were significant factors. ”

---

## [Editor Report · Decision Letter 2]

4 Apr 2025

PONE-D-24-23927R2Effect of Inverted Visual Acceleration Profile on Vestibular Heading Perception.PLOS ONE

Dear Dr. Crane,

Thank you for submitting your manuscript to PLOS ONE. After careful consideration, we feel that it has merit but does not fully meet PLOS ONE’s publication criteria as it currently stands. Therefore, we invite you to submit a revised version of the manuscript that addresses the points raised during the review process.

After assessing your revisions, I feel that Reviewer #4's comment regarding the ANOVA design has not been fully resolved. Reviewer #4 rightly pointed out that for a 3x2 repeated-measures ANOVA design with n=20 participants, the degrees of freedom for the two-level factor main effect are F(1,19), and for the three-level factor main effect as well as for the interaction effect the df are F(2,38). Given that you report F(1,2) as the df of the two-way ANOVA, it appears that statistics were not properly run on individual means. Moreover, reporting of ANOVA and t-test results still does not follow the conventional form pointed out by Reviewer #4 (i.e., "F(df1,df2) = F-value, p = p-value" for each ANOVA effect and "t(df) = t-value, p = p-value" for each t-test). Please address these issues with your next revision.

We look forward to receiving your revised manuscript.

Kind regards,

Patrick Bruns

Academic Editor

PLOS ONE
---

## [Author Response · Author response to Decision Letter 3]

5 Apr 2025

After assessing your revisions, I feel that Reviewer #4's comment regarding the ANOVA design has not been fully resolved. Reviewer #4 rightly pointed out that for a 3x2 repeated-measures ANOVA design with n=20 participants, the degrees of freedom for the two-level factor main effect are F(1,19), and for the three-level factor main effect as well as for the interaction effect the df are F(2,38). Given that you report F(1,2) as the df of the two-way ANOVA, it appears that statistics were not properly run on individual means. Moreover, reporting of ANOVA and t-test results still does not follow the conventional form pointed out by Reviewer #4 (i.e., "F(df1,df2) = F-value, p = p-value" for each ANOVA effect and "t(df) = t-value, p = p-value" for each t-test). Please address these issues with your next revision.

To review here are the previous comments of reviewer #4:

Reviewer #4: The authors have addressed most of my comments but I just want to clarify the ANOVA and how it has been reported.It is a bit confusing as in the analysis the ANOVA is given as an offset x stimulus 3(45 deg, 60 deg and 70 deg)x2(normal and inverted).But in the results it seems to be 6x1 (45 deg x normal, 60 deg x normal, 70 deg x normal, 45 deg x inverted, 60 deg x inverted and 70 deg x inverted).If it is a 2x3 the DoF should be:for the stimulus 1,19for the offset 2,38for the interaction 2,38If it is a 6x1 then the DoF is 5 but it does not test the offset and stimulus separately, which means was a followup ANOVA conducted.Can the authors clarify this.What I would suggest is for the ANOVAs report the results as F(df1,df2)=?,p< and for the t-tests state t(df)=??,p

Response: We have made the proposed changes:

The first paragraph of the results now states:

A three-way ANOVA was performed using visual stimulus type (normal, inverted) and the three non-zero offsets (45, 60, 75°). This (ANOVA, F(1,38) = 9.61) revealed that visual stimulus type is the only statistically significant factor (p = 0.0022) with no effect of offset magnitude (F(2, 38) = 0.28, p = 0.76). There was a significant effect between subjects (ANOVA, F(6,19) = 13.2, p < 0.0001) A paired T-test of inverted vs. normal visual stimulus was statistically significant t(118) = 6.487, p < 0.0001 (two-tailed) when all the non-zero offsets were considered. When the offsets were considered separately, the effect remained statistically significant at 45° t(39) = 3.04, p = 0.004, 60° t(39) = 4.18, p = 0.0002, and 75° t(39) = 4.58, p < 0.0001.

The third paragraph is also changed:

When the potential effects of visual stimulus type and offset were examined there were no statistically significant effects using a three-way ANOVA to look at stimulus type and non-zero offsets. When the visual stimulus profile was examined (ANOVA, F(1,38) = 0.36) revealed that the stimulus type (p = 0.55) was not a significant factor. When offset magnitude was examined (ANOVA. F(2,38) = 0.20) there was also no significant effect (p = 0.82). As with the bias, there was significant variation in sigma between subjects (ANOVA, F(6,19) = 3.38, p < 0.0001). Overall, the normal stimulus had a sigma of 6.1 ± 0.4 (SEM) and the inverted stimulus had a sigma of 5.6 ± 0.4 (SEM), but this difference was not statistically significant t(179) = -0.94, p = 0.35.

---

## [Editor Report · Decision Letter 3]

8 Apr 2025

Effect of Inverted Visual Acceleration Profile on Vestibular Heading Perception.

PONE-D-24-23927R3

Dear Dr. Crane,

We’re pleased to inform you that your manuscript has been judged scientifically suitable for publication and will be formally accepted for publication once it meets all outstanding technical requirements.

Kind regards,

Patrick Bruns

Academic Editor

PLOS ONE

Additional Editor Comments (optional):

Thank you for addressing the remaining issues. You might want to double-check the following points for correctness at the proof stage:

"three-way ANOVA": shouldn't this be "two-way ANOVA" as there seem to be only two factors?

ANOVA df for main effect of factor visual stimulus type: shouldn't this be F(1,19) rather than F(1,38)?

The modified sentence on p. 12 ("When the visual stimulus profile was examined...revealed that the stimulus type") appears grammatically incorrect.

Are the df for the t-test reported on p. 12 ("t(179)") correct?
---

## [Editor Report · Acceptance letter]

PONE-D-24-23927R3

PLOS ONE

Dear Dr. Crane,

I'm pleased to inform you that your manuscript has been deemed suitable for publication in PLOS ONE. Congratulations! Your manuscript is now being handed over to our production team.

Kind regards,

on behalf of

Dr. Patrick Bruns

Academic Editor

PLOS ONE